# Remote Sensing-Based Classification of Winter Irrigation Fields Using the Random Forest Algorithm and GF-1 Data: A Case Study of Jinzhong Basin, North China

Qiaomei Su [1], Jin Lv [2,3], Jinlong Fan [2,*], Weili Zeng [1,2], Rong Pan [1,2], Yuejiao Liao [1,2], Ying Song [1,2], Chunliang Zhao [2,4], Zhihao Qin [4,5] and Pierre Defourny [6]

[1] Department of Surveying and Mapping, College of Mining Engineering, Taiyuan University of Technology, Taiyuan 030002, China; suqiaomei@tyut.edu.cn (Q.S.); cengweili0835@link.tyut.edu.cn (W.Z.)
[2] National Satellite Meteorological Center, Beijing 100081, China; zhaochunliang@caas.cn (C.Z.)
[3] School of Public Administration, China University of Geosciences, Wuhan 430074, China
[4] MOA Key Laboratory of Agricultural Remote Sensing, Institute of Agro-Resources and Regional Planning, Chinese Academy of Agricultural Sciences, Beijing 100081, China; qinzhihao@caas.cn
[5] School of Geographic Science and Planning, Nanning Normal University, Nanning 530100, China
[6] Earth and Life Institute, Université Catholique de Louvain, 1348 Louvain-la-Neuve, Belgium; pierre.defourny@uclouvain.be
[*] Correspondence: fanjl@cma.gov.cn

**Abstract:** Irrigation is one of the key agricultural management practices of crop cultivation in the world. Irrigation practice is traceable on satellite images. Most irrigated area mapping methods were developed based on time series of NDVI or backscatter coefficient within the growing season. However, it has been found that winter irrigation out of growing season is also dominating in north China. This kind of irrigation aims to increase the soil moisture for coping with spring drought and reduce the wind erosion in spring. This study developed a remote sensing-based classification approach to identify irrigated fields out of growing season with Radom Forest algorithm. Four spectral bands and all Normalized Difference Vegetation Index (NDVI) like indices computed from any two of these four bands for each of the seven scenes of GF-1 satellite data were used as the input features in the building of separated RF models and in applying the built models for the classification. The results showed that the mean of the highest out-of-bag accuracies for seven RF models was 94.9% and the mean of the averaged out-of-bag accuracies in the plateau for seven RF models was 94.1%; the overall accuracy for all seven classified outputs was in the range of 86.8–92.5%, Kappa in the range of 84.0–91.0% and F1-Score in the range of 82.1–90.1%. These results showed that the classification was neither overperformed nor underperformed as the accuracies of all classified images were lower than the model ones. This study also found that irrigation started to be applied as early as in November and irrigated fields were increased and suspended in December and January due to freezing conditions. The newly irrigated fields were found again in March and April when the temperature rose above zero degrees. The area of irrigated fields in the study area were increasing over time with sizes of 98.6, 166.9, 208.0, 292.8, 538.0, 623.1, 653.8 km$^2$ from December to April, accounting for 6.1%, 10.4%, 12.9%, 18.2%, 33.4%, 38.7%, and 40.6% of the total irrigatable land in the study area, respectively. The results showed that the method developed in this study performed well. This study found on the satellite images that 40.6% of irrigatable fields were already irrigated before the sowing season and the irrigation authorities were supposed to improve their water supply capacity in the whole year with this information. This study may complement the traditional consideration of retrieving irrigation maps only in growing season with remote sensing images for a large area.

**Keywords:** irrigation map; irrigation fields; classification; GF-1

## 1. Introduction

Irrigation is one of the key agricultural management practices of crop cultivation in the world [1–4]. Irrigation reduces adverse effects of drought, increases crop yield, and finally maintains a good agricultural production profit. Irrigation consumes a lot of water resources and thus efficient water use management requires timely irrigation information in large regions [5]. Irrigated crop land, irrigation events and irrigation water amount are provide important information in the support of sustainable water resource management. Studies on hydrology [6], water availability and water use [7], and their interaction with agricultural production and food security [8], all require accurate information on the location and extent of irrigated croplands. Detailed knowledge about the timing and the amounts of water used for irrigation over large areas [3] is also of importance for various studies and applications.

Irrigation practice is traceable on satellite images [9]. A few global irrigation maps such as the Global Map of Irrigated Areas (GMIAs) [10] and the Global Irrigated Area Map (GIAM) [11] have become available. Recently, Wu [12] retrieved a 30-m resolution global maximum irrigation extent (GMIE) using the Normalized Difference Vegetation Index (NDVI) and NDVI deviation (NDVIdev) thresholds in the dry and driest months. Zajac [13] derived the European Irrigation Map for the year 2010 (EIM2010) underpinned by the agricultural census data. Siddiqui [14] developed irrigated area maps for Asia and Africa regions using canonical correlation analysis and time lagged regression at 250 m resolution for the year 2000 and 2010. Zhang [15] produced annual 500-m irrigated cropland maps across China for 2000–2019, using a two-step strategy that integrated statistics, remote sensing, and existing irrigation products into a hybrid irrigation dataset. Zhao [16] developed crop class based irrigated area maps for India using net sown area and extent of irrigated crops from the census and land use land cover data at 500 m spatial resolution for the year 2005. Ambika [17] developed annual irrigated area maps at a spatial resolution of 250 m for the period of 2000–2015 using data from the Moderate Resolution Imaging Spectroradiometer (MODIS) and 56 m high-resolution land use land cover (LULC) information in India. Gumma [18] mapped irrigated agricultural areas for Ghana using remote-sensing methods and protocols with a fusion of 30 m and 250 m spatial resolution remote-sensing data. Xie [19] mapped the extent of irrigated croplands across the conterminous U.S. (CONUS) for each year in the period of 1997–2017 at 30 m resolution, using the generated samples along with remote sensing features and environmental variables to train county-stratified random forest classifiers annually.

Most irrigated area mapping methods above-mentioned were based on time series of NDVI at a relatively low resolution of 250–1000 m. Disaggregating statistics data on the grid is another way to generate the irrigation maps. For example, the European irrigation map (EIM) [20] was created by disaggregating regional-level statistics on irrigated cropland areas into a 100 × 100 m grid, using a land cover map and constrained by the Global Map of Irrigated Areas (GMIAs) [10]. The remote sensing-based classification approach is also a great way to produce the irrigated crop maps. Salmon [21] used supervised classification of remote sensing, climate, and agricultural inventory data to generate a global map of irrigated, rain-fed, and paddy croplands. Lu [22] tried to use pixel-based random forest to map irrigated areas based on two scenes of GF-1 satellite images at 16 m in an irrigated district of China, during the winter-spring irrigation period of 2018. Magidi [23] developed a cultivated areas dataset with the Google Earth Engine (GEE) and further used the NDVI to distinguish between irrigated and rainfed areas. A large variety of classification methods at different scales and showing various levels of accuracy can be found in the literature [24–31]. Many applications and the tool of cloud-based and open source in classification have been developed recently [32,33]. However, the cloudy contamination and revisit time of optical satellite creates a major limitation to accurately identifying irrigation signature on the imagery. SAR imagery is less impacted by the cloud and has the advantage of building a long time series data to detect the irrigation signature. A number of studies [34–36] used timer series of SAR images to detect the irrigation event. The fusion of optical and SAR time

series images for classification is also progressing well in recent years [37–39] in order to reduce the cloudy issue on the optical image. One study [40] assessed the value of satellite soil moisture for estimating irrigation timing and water amounts.

All these above-mentioned studies were designed to identify the irrigation signature mainly in growing season as crop develops. However, irrigation also happens out of season due to various reasons, such as sufficient water supply out of season, cheaper water prices, and lower energy prices as well as manpower availability. This kind of irrigation practice should be given more attention as the winter irrigation is dominating in this region. Therefore, this study aimed to develop a method to identify the irrigated fields and help irrigation authorities know the irrigation situation before the growing season arrives to improve their water supply capacity in the whole year so that the crop production may be stably maintained. In this study area, it found a great number of fields already irrigated in winter and in early spring, although fields are bare soil and large volumes of irrigation water was applied to the fields. This kind of irrigation practice aims to keep enough soil moisture for sowing crops at the beginning of growing season in spring to avoid irrigation water competition and in preparation for coping with spring drought. This case also complements the consideration from those researchers who are developing irrigation maps within growing season for a large area or at a global level.

## 2. Study Area and Data

### 2.1. Study Area

The study area is located in the midstream of the Fen River, at the center of Shanxi province in north China (Figure 1). This region, also known as the Jinzhong basin, spans approximately 150 km in length and 30–40 km in width, covering a total area of approximately 5000 km$^2$. Three prominent rivers—Fen River, Wenyu River, and Xiao River—grace the landscape. The practice of irrigation has deep historical roots here, spanning over a millennium. The irrigation domain of the Fen River covers 1046.1 km$^2$ of arable land, benefiting three cities and a vast agricultural community of a million farmers [41]. The Wenyu River, a tributary of the Fen River, irrigates an area of 341.8 km$^2$ of arable land [42]. Similarly, the Xiao River irrigates an area of 221.7 km$^2$ of arable land [43]. These irrigation facilities remain integral, with flooding irrigation still prevailing through the irrigating channels that nourish the fields. The study area holds prominence as a key grain production hub within Shanxi province, significantly contributing to regional food security. Its agricultural landscape is diverse, incorporating staple crops such as maize, sorghum, and winter wheat. Orchards, vegetable greenhouses, and other crop fields further enrich its agricultural mosaic. The conventional growing season extends from May to September, yet this area supports winter wheat cultivation throughout the winter months. Planting commences from early to mid-October. After the winter wheat harvest, short-lived crops are sown to evade early autumn frost. Presently, a few fields are dedicated to winter wheat cultivation, while most fields remain fallow during winter. This has led to the application of winter irrigation to these fallow fields. Summer witnesses the cultivation of maize in most fields, a crop that particularly benefits from winter irrigation. Climatically, the study area falls within the temperate continental seasonal climate zone, experiencing distinct seasons—spring, summer, autumn, and winter. Notably, winter and spring receive less rainfall compared to the pronounced rainy season during summer.

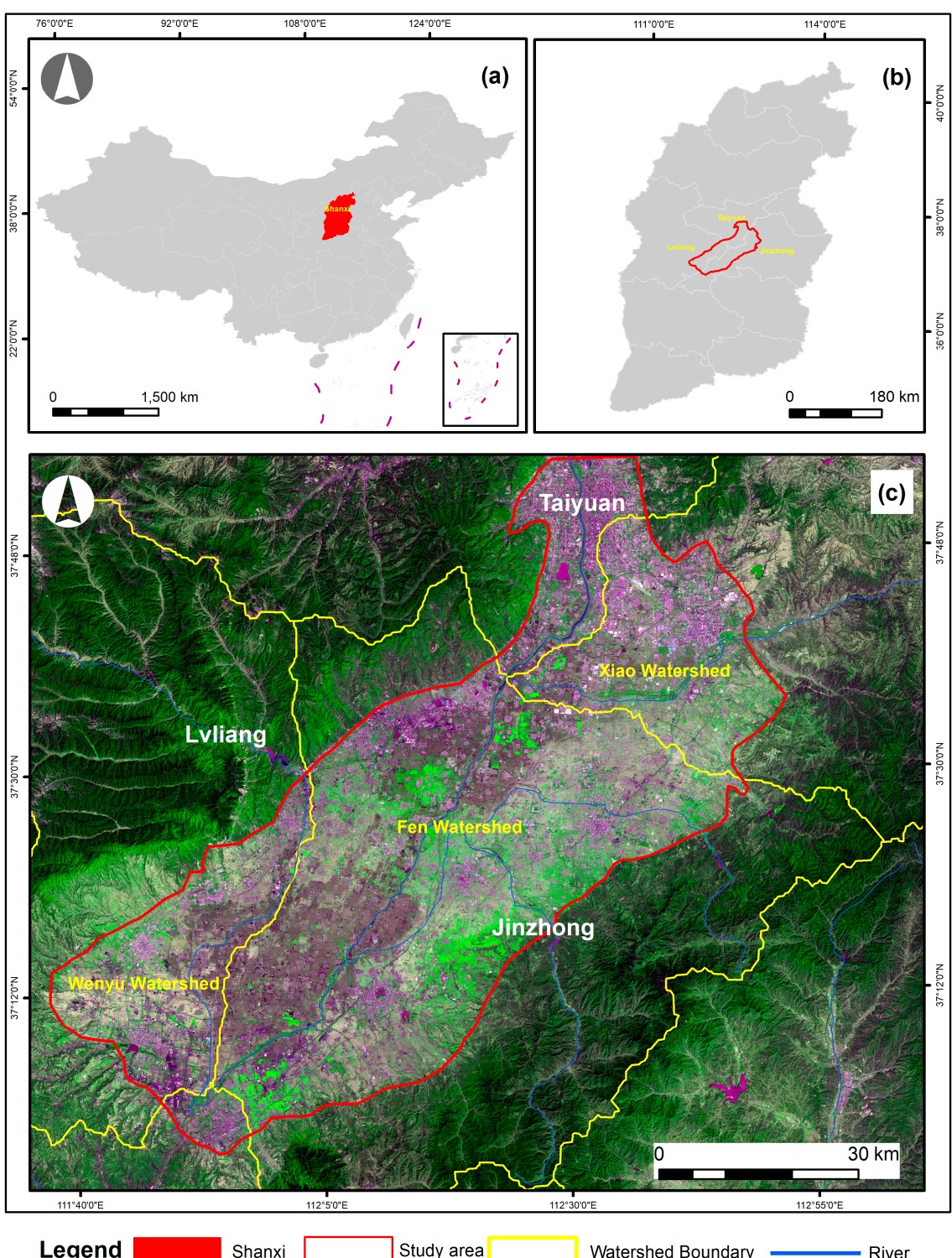

**Figure 1.** The location of the study area: (**a**) shows the Shanxi province in China, (**b**) shows the study area in Shanxi province, and (**c**) shows the study area illustrated on the GF-1 image on 29 April 2023, respectively.

### 2.2. Satellite Data and Processing

GF, the acronym of Gaofen in Chinese and high resolution in English, is one of the key Earth observation programs in China. As the first satellite of the Chinese High Resolution Earth Observation System, GF1 Satellite was successfully launched on 26 April 2013 [44]. Four sets of multiple spectral cameras (wide field of view, WFV) were equipped onboard

GF-1 and had a mosaic coverage spanning 800 km at 16-m spatial resolution and a 4-day revisit frequency [45]. As one of limitations in comparison with other high resolution satellite imagery, WFV has only four bands listed in Table 1. The L1B data of GF-1 WFV data in this study were collected from National Satellite Meteorological Center, China. After the visual check of all images, the images with less than 25% cloud coverage were selected and processed for this study. Thereafter, the FLAASH approach was used to perform the atmospheric correction [46]. The RPC Orthorectification approach [46] was used to perform the geometrical correction from L1B data. Considering that GF-1 had relatively large geometric errors, a 10 m Sentinel-2B image obtained on 17 October 2022 was used as the reference image to co-register all GF-1 images with the image chip matching method. The results for the co-registration of all GF-1 images will be reported in another article in preparation. The images in the same day were mosaiced and tailored to the study area. Finally, the available images were listed in Table 2. Due to partial cloud contamination, the satellite data in November and February were removed and the final valid data fit in seven dates. In order to make it compatible with other high resolution satellite data, like Sentinel-2 and Landsat 8/9, the spatial resolution of GF-1 WFV in this study was set to 15 m, not 16 m as expected normally.

**Table 1.** Band Specification and Spatial Resolution of GF-1 WFV.

| Band Number | Central Wavelength (nm) | Bandwidth (nm) | Resolution (m) |
| --- | --- | --- | --- |
| 1 | 485 | 70 | 16 |
| 2 | 555 | 70 | 16 |
| 3 | 660 | 60 | 16 |
| 4 | 830 | 120 | 16 |

**Table 2.** The used GF-1 WFV data.

| No. | 1 | 2 | 3 | 4 | 5 | 6 | 7 |
| --- | --- | --- | --- | --- | --- | --- | --- |
| Date | 27 December 2022 | 4 January 2023 | 25 January 2023 | 3 March 2023 | 27 March 2023 | 8 April 2023 | 29 April 2023 |

### 2.3. Field Data and Training Samples

A 3-day field campaign was carried out on 24–26 February 2023. During the field campaign, the georeferenced pictures were taken with a GPS camera along the roads following predefined itineraries in the study area. At home, the land cover classes with the longitude and latitude coordinates were retrieved by visually screening pictures with the tool developed for the photo data interpretation [47–49]. During the field campaign, irrigated fields were partially frozen and waterlogged and it was also easy to identify on the satellite images. The final output of this process was a formatted file gathering all GPS points with corresponding classes, class codes, author, roadside (left or right), collecting dates, and times and the corresponding picture file names. Finally, 3616 ground truth pictures were valid and with spatial reference. All those sample points were distributed over the study area as shown in Figure 2.

The field samples include built-up, water body, tree, orchards, irrigated filed, bare land, winter wheat, green house, and others. These samples are point-based ground truth and not ideally and evenly distributed in the study area. These field samples were used for further collecting, more and well-distributed training and validation samples by visually interpreting satellite images. The final samples were randomly separated into two groups with a ratio of 70% to 30%. 70%of the samples were used to build the classification model and perform the classification, while 30% were used for the validation of classified images. Following our previous experiences [47–49], the distance between two samples was taken into account in the sample separation process. In case both were too close, all pixels in the adjacent area were chosen as either training or validation. For instance, all samples

at the level of image pixel taken in one field represented only one class, so it is good to treat them as one big sample. The threshold of the distance in this study was set as 900 m. This step avoids the strong spatial correlation among samples. Table 3 lists the description of all classes identified for final classification. Table 4 lists the number of samples and the proportion for each class at 15-m level for this study. Irrigation 1 represents the fields waterlogged or frozen in winter after the large volume flooding irrigation. Irrigation 2 represents the fields with the high soil moisture but without surface water. Two conditions explicated represented water amount difference in the fields. Irrigation 1 meant there was too much water in the fields. Due to a large volume of water applied to the field and weak evaporation in winter, no classification samples for Irrigation 2 were identified on 17 December, 4 January, and 25 January. In the other dates, the two kinds of irrigation conditions in the field were able to be identified.

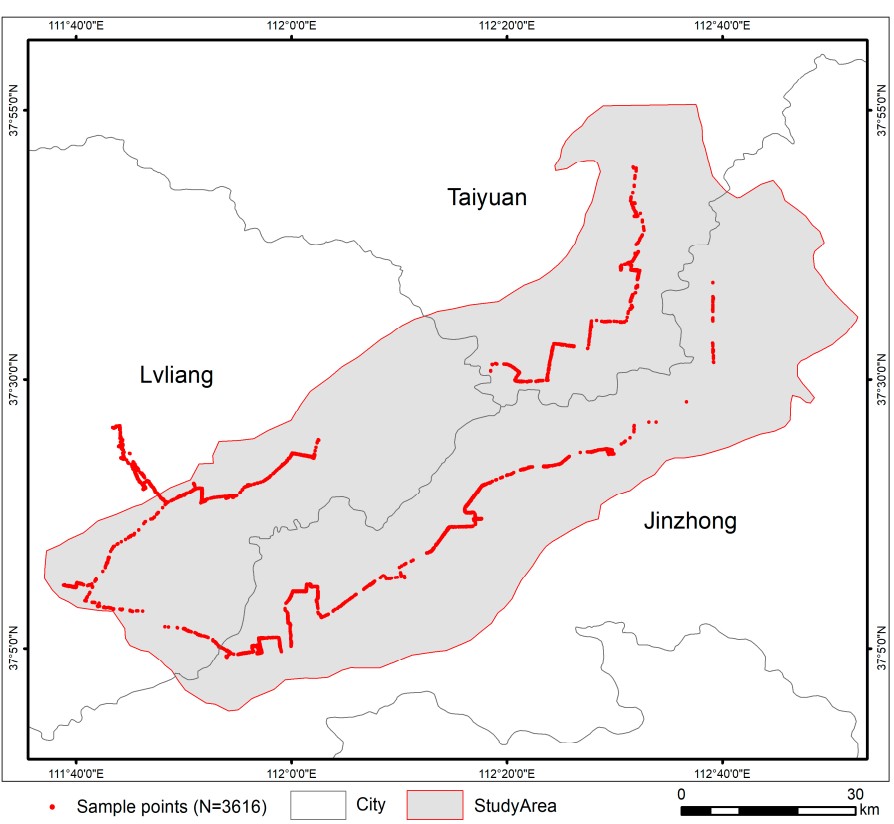

**Figure 2.** The distribution of field samples in the study area.

**Table 3.** Land cover types and their brief descriptions.

|  | Class | Acronym | Description |
|---|---|---|---|
| Cropland | Irrigation 1 | I1 | Waterlogged or Frozen field after irrigation |
|  | Irrigation 2 | I2 | Field with high soil moisture after irrigation |
|  | Winter Wheat | WW | Winter wheat field |
|  | Straw Covered Cropland | SC | Cropland covered by the straw or other residues out of season |
|  | Bare Cropland | BC | Bare and no covered cropland out of season |
|  | Greenhouse | GH | Greenhouse for vegetable or other cash crops |
|  | Orchards | OC | Fruit trees plantation |
|  | Plantation | PL | Cropland planted with wood or shrub trees |
| Non Cropland | Built-up | BU | Artificial area including building, road, and factory |
|  | Barren Land | BL | No vegetation covered area in rock mountain |
|  | Deciduous Forest | DF | Deciduous tree and shrub |
|  | Evergreen Forest | EF | Coniferous tree and shrub |
|  | Water Body | WB | Lake, River, Dam, and other Water body |

**Table 4.** The number of training samples and the proportion for each class.

| DATE\Class | 27 December 2022 | | 4 January 2023 | | 25 January 2023 | | 3 March 2023 | | 27 March 2023 | | 8 April 2023 | | 29 April 2023 | |
|---|---|---|---|---|---|---|---|---|---|---|---|---|---|---|
| | No. (Pixel Counts) | Proportion (%) | No. (Pixel Counts) | Proportion (%) | No. (Pixel Counts) | Proportion (%) | No. (Pixel Counts) | Proportion (%) | No. (Pixel Counts) | Proportion (%) | No. (Pixel Counts) | Proportion (%) | No. (Pixel Counts) | Proportion (%) |
| BC | 1898 | 9.2 | 1705 | 7.9 | 1604 | 7.8 | 1466 | 7.1 | 1602 | 8.3 | 1569 | 7.8 | 1485 | 7.7 |
| BL | 1071 | 5.2 | 1015 | 4.7 | 1045 | 5.1 | 1085 | 5.2 | 1052 | 5.5 | 957 | 4.8 | 1014 | 5.3 |
| BU | 7349 | 35.6 | 6836 | 31.8 | 7662 | 37.3 | 7502 | 36.1 | 6873 | 35.6 | 7403 | 36.8 | 7077 | 36.8 |
| DF | 1466 | 7.1 | 1214 | 5.6 | 1339 | 6.5 | 1402 | 6.7 | 1319 | 6.8 | 1347 | 6.7 | 1259 | 6.5 |
| EF | 2067 | 10 | 2016 | 9.4 | 2067 | 10.1 | 2089 | 10 | 2139 | 11.1 | 1801 | 8.9 | 2060 | 10.7 |
| GH | 704 | 3.4 | 2561 | 11.9 | 695 | 3.4 | 745 | 3.6 | 687 | 3.6 | 514 | 2.6 | 700 | 3.6 |
| I1 | 1588 | 7.7 | 1523 | 7.1 | 1562 | 7.6 | 879 | 4.2 | 812 | 4.2 | 459 | 2.3 | 371 | 1.9 |
| I2 | 0 | 0 | 0 | 0 | 0 | 0 | 1327 | 6.4 | 1408 | 7.3 | 2587 | 12.8 | 1461 | 7.6 |
| OC | 342 | 1.7 | 436 | 2.1 | 402 | 2 | 347 | 1.7 | 374 | 1.9 | 399 | 2 | 531 | 2.8 |
| PL | 121 | 0.6 | 204 | 0.9 | 108 | 0.5 | 167 | 0.8 | 158 | 0.8 | 112 | 0.6 | 118 | 0.6 |
| SC | 1111 | 5.4 | 1051 | 4.9 | 829 | 4 | 770 | 3.7 | 391 | 2 | 164 | 0.8 | 155 | 0.8 |
| WB | 2278 | 11 | 2407 | 11.2 | 2114 | 10.3 | 2412 | 11.6 | 1806 | 9.4 | 1968 | 9.7 | 2140 | 11.2 |
| WW | 622 | 3.1 | 546 | 2.5 | 623 | 3 | 602 | 2.9 | 674 | 3.5 | 858 | 4.2 | 879 | 4.5 |
| Total | 20,617 | 100 | 21,514 | 100 | 20,544 | 100 | 20,793 | 100 | 19,295 | 100 | 20,138 | 100 | 19,250 | 100 |

## 3. Methodology

### 3.1. Classification Flowchart

Figure 3 presents the flowchart for this study. The GF-1 WFV satellite data from November 2022 to April 2023 were collected and processed. The main processing steps include calibration, geometric, and atmospheric correction. The calibration coefficients are available from the web portal [50]. The DN values in the images were converted to reflectance in the calibration step. The RPC Orthorectification approach [46] was applied to do geometric correction of all GF-1 L1b data. Thereafter, the FLAASH approach [46] was used to perform the atmospheric correction. In order to make all GF-1 images geometrically match each other, all images were co-registered with one scene of 10 m Sentinel-2 image obtained on 17 October 2022. Then, all finely co-registered images were mosaiced based on observing date and tailored to the study area.

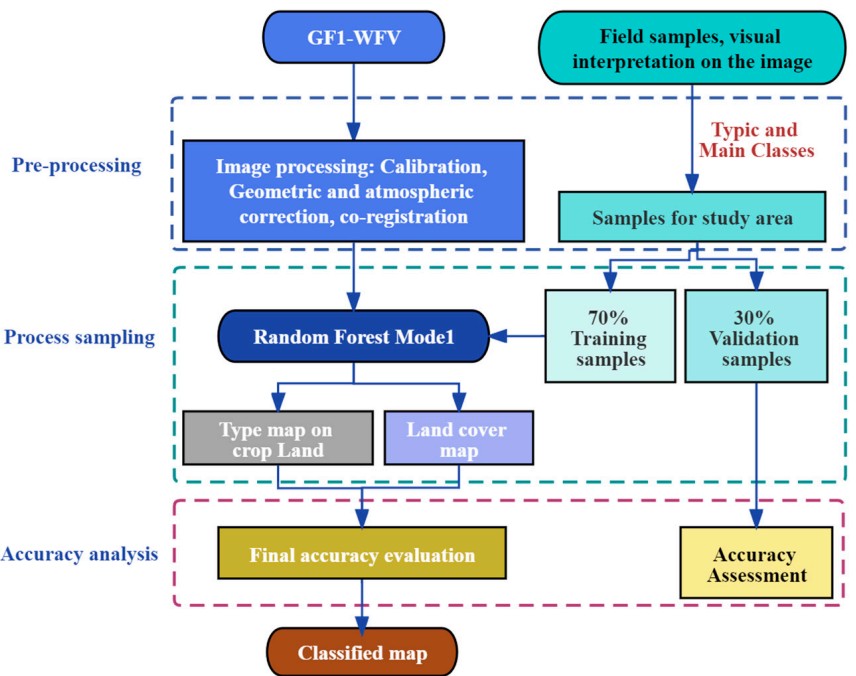

**Figure 3.** The flowchart for this study.

At the same period, the field data were collected. In general, these field data were not enough for the classification in terms of spatial distribution and statistical amount. In order to make well-spatially distributed training samples, these geotagged photos were linked with the satellite image to help skilled interpreters to visually identify more samples on satellite images. The classification samples corresponding to each image were separated into training and validation sets by a ratio of 70% to 30%.

In the next step, a Random Forest classifier was used to execute the classification with each image and corresponding training samples. The classified image was checked by validation samples based on error confusion matrix and expert knowledge visually. If the classification accuracy is not acceptable, tuning the training samples may improve the classification quality. Referring to F1-Score for each type, the classification samples were further tuned by spatially increasing or removal of some bad quality samples until the result was acceptable. Once the accuracy is acceptable, the final classified map is output, and the final accuracy is reported. Considering the irrigation changes over time, the training samples and validation samples were collected separately based on each image. The classification was carried out one image by one image and not worked with time series [51,52]. When all classified images were done, the final maps and statistics were made for further analysis.

### 3.2. Classifier Algorithm

The supervised classification algorithm is widely used at present. In supervised classification, the training samples must provide an association with the input images. The final class for each pixel is decided by the classifier. Many literatures [33,47,51,52] has proved that the accuracy from Random Forest (RF) often overperforms other supervised classifiers, e.g., Maximum Likelihood (ML) and Support Vector Machine (SVM). RF has become the popular classifier in recent years as it is robust and easy to apply and only few parameters need to be set and tuned accordingly. Therefore, RF was selected for this study. RF is a supervised machine learning algorithm and a kind of ensemble of the decision trees. RF can handle high dimension of and redundant input satellite data and does not have preference to the certain satellite data. What is of importance in executing RF is that it has to pay attention to the overfitting of the classification model. The detailed algorithm of RF may refer to the literature [53–57]. RF has only two key parameters to be considered. One is the number of features and another is the number of trees. RF will use the certain number of randomly selected features to build the model. It is not the case that the higher the accuracy, the more features are used. The highest accuracy may be reached with only a contained number of features. The drawback is that the higher number of features increases computing time. In this study, the number of features was set as the square root of the number of input bands of the image. The accuracy will reach the plateau after the certain number of the tree and there is no need to set a very high number. After the tests, 100 was set for the number of the tree. More features may increase the accuracy of the classified image. Every two spectral bands may be used to calculate a NDVI like index following our previous study [49]. So, in this study, all possible NDVI like indices were calculated and added with four spectral bands as the input features for the final classification.

### 3.3. Validation Methods

The error confusion matrix is usually used to quantitatively evaluate the accuracy of the classified image. The overall accuracy, OA, the Kappa, and F1-Scores may be further calculated based on the error confusion matrix. This study used above-mentioned indices to evaluate the accuracy. The formulars of computing OA, the Kappa, and F1-Scores may refer to the literatures [47–49].

In this study, the validation of each classified images was carried out separately with its independent validation samples. The statistic was computed by counting the number of pixels for each class.

## 4. Results and Analysis

### 4.1. The Classification Model Accuracy Analysis

The accuracy of the classified model determines the top boundary of accuracy that classification may reasonably reach. The higher the model accuracy goes, the higher the classification accuracy may reach. Figure 4 shows the accuracies of out-of-bag of all models of Random Forest algorithm. The accuracy of out-of-bag is increasing as the number of trees increases and the accuracy reaches the plateau after 30 tries. In this study, it should be reasonable as the number of trees was set to 100 according to Figure 4. There were slight differences among all these models but the difference was in a range of about 2% that meant it was quite small. Figure 5 shows the averaged value and maximum value of model accuracy in the range of plateau taken from 50 to100 in this study. The averaged highest accuracy for seven models was 94.9% and the averaged mean accuracy was 94.1%. These data show that all models were good and acceptable.

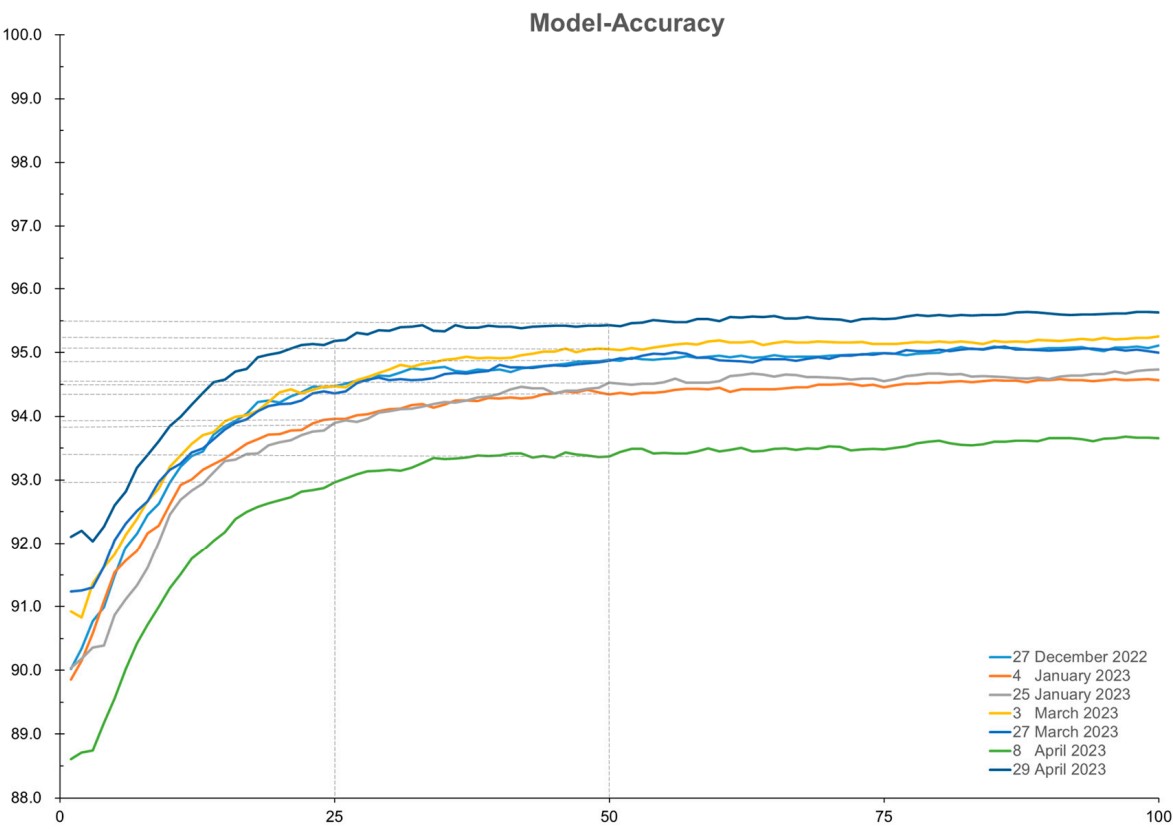

**Figure 4.** The accuracies of models corresponding to each image.

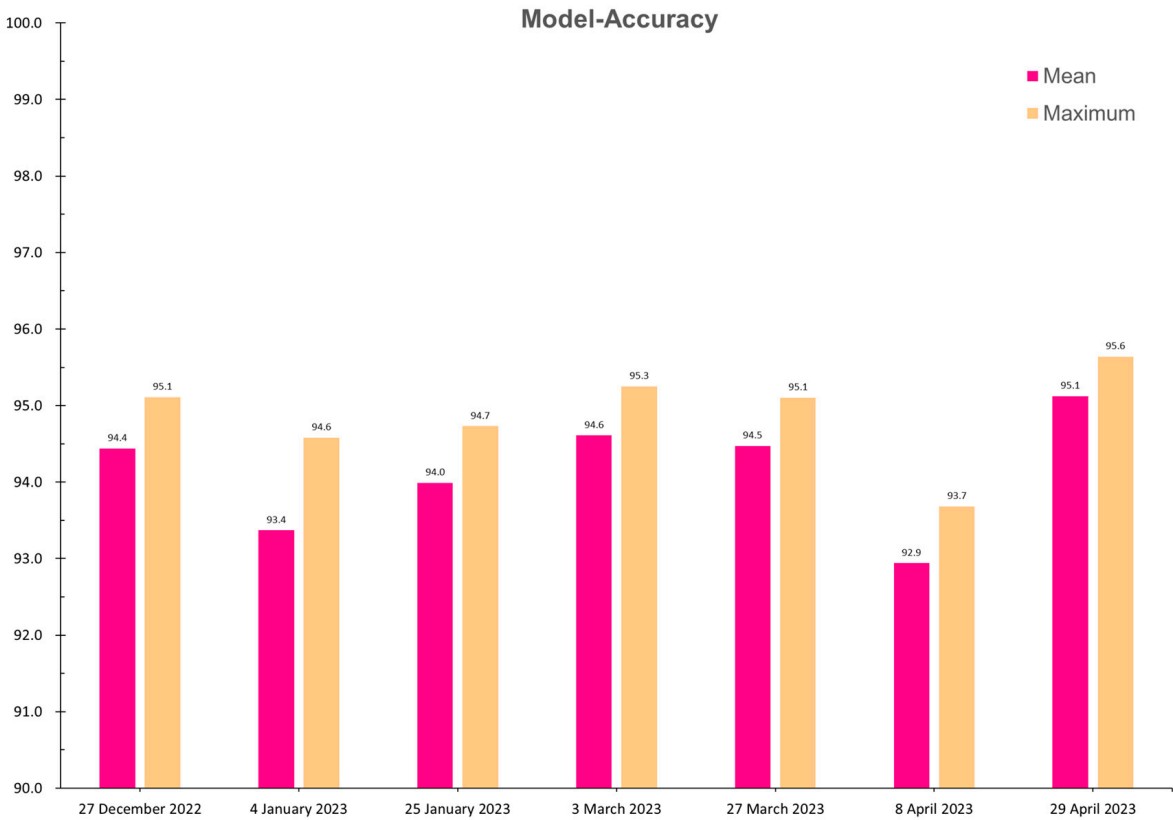

**Figure 5.** The maximum and mean accuracies of models corresponding to each image in the plateau.

### 4.2. The Classified Images and Accuracies

GF-1 WFV has only four bands, namely blue, green, red, and near infrared. These bands are the basic features that may be used for the classification. However, according to our previous study [49], every two spectral bands of four bands were used to calculate a NDVI like index and added as the input features. Figure 6 shows the classified images for all seven dates. The accuracies are listed in Table 5.

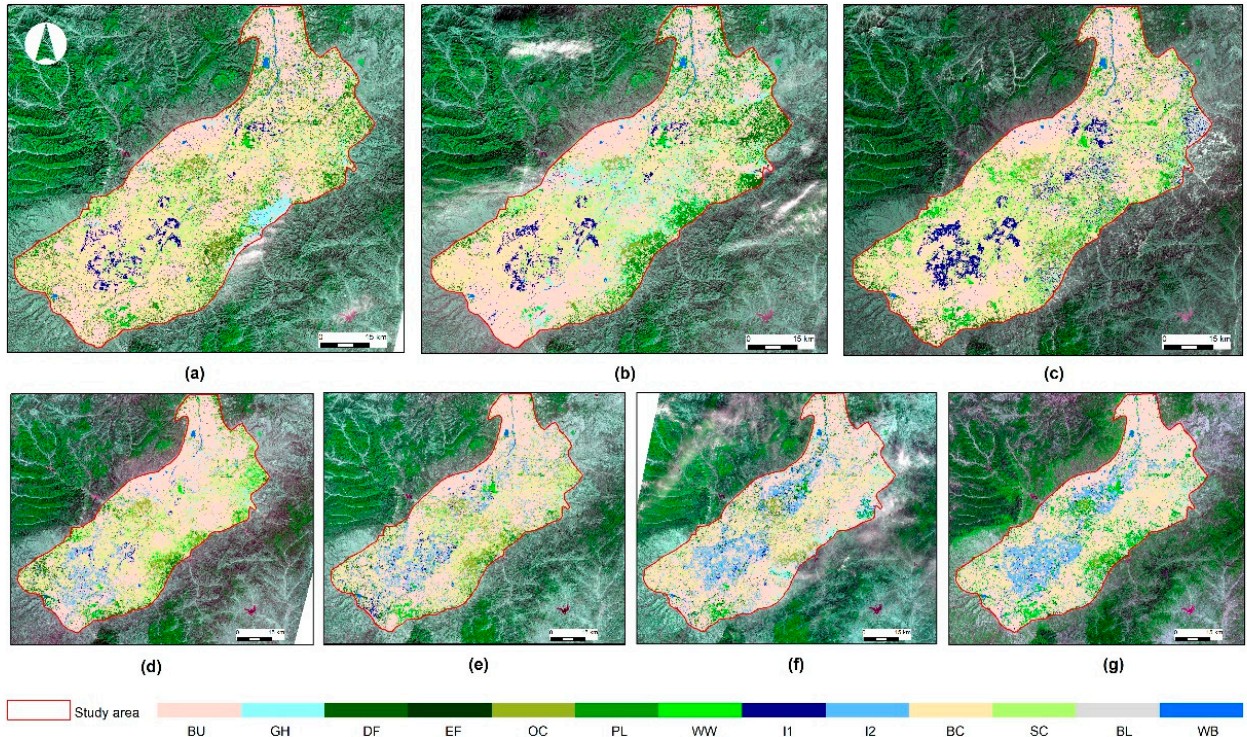

**Figure 6.** The classified images: (**a**) on 27 December 2022, (**b**) on 4 January 2023, (**c**) on 25 January 2023, (**d**) on 3 March 2023, (**e**) on 27 March 2023, (**f**) on 8 April 2023, (**g**) on 29 April 2023, respectively.

**Table 5.** Accuracies for all dates.

|  | 27 December 2022 | 4 January 2023 | 25 January 2023 | 3 March 2023 | 27 March 2023 | 8 April 2023 | 29 April 2023 |
|---|---|---|---|---|---|---|---|
| OA | 92.5 | 90.3 | 89.8 | 88.8 | 91.1 | 86.8 | 90.9 |
| Kappa | 91.0 | 88.0 | 88.0 | 87.0 | 89.0 | 84.0 | 89.0 |
| F1-Score | 90.1 | 85.9 | 84.2 | 83.7 | 86.7 | 82.1 | 86.5 |
| I1 | 97.2 | 97.1 | 92.2 | 88.1 | 87.2 | 91.7 | 86.0 |
| I2 | - | - | - | 72.7 | 75.6 | 80.3 | 95.8 |
| WW | 93.4 | 75.9 | 73.0 | 82.9 | 97.6 | 83.5 | 95.1 |
| SC | 85.7 | 79.3 | 81.4 | 59.1 | 61.6 | 55.4 | 63.3 |
| BC | 83.4 | 75.4 | 73.2 | 60.5 | 74.0 | 82.4 | 86.3 |
| GH | 91.9 | 87.5 | 81.6 | 96.1 | 96.7 | 81.0 | 87.2 |
| OC | 86.8 | 93.7 | 73.4 | 87.2 | 82.2 | 83.2 | 96.1 |
| PT | 77.1 | 63.0 | 63.7 | 77.5 | 71.7 | 39.2 | 68.2 |
| BU | 94.1 | 92.0 | 93.4 | 92.4 | 93.6 | 90.3 | 93.7 |
| BL | 89.7 | 91.5 | 95.5 | 86.4 | 95.3 | 59.3 | 86.3 |
| DF | 85.8 | 82.0 | 95.4 | 89.6 | 94.3 | 88.7 | 82.4 |
| EF | 98.1 | 98.3 | 99.2 | 98.4 | 99.4 | 95.1 | 91.9 |
| WB | 97.5 | 95.0 | 95.5 | 97.0 | 97.3 | 93.8 | 93.3 |

According to Table 5, in terms of overall classification performance, the OA for 7 classifications was between 86.8 and 92.5, Kappa between 84.0 and 91.0, F1-Score between 82.1 and 90.1. After the visual check of all classified images and looking at all these overall performance accuracy indictors, it concluded that these classifications were well-performed. As irrigation is the focus for this study, in the training phase, two kinds of irrigation condi-

tions were identified. Irrigation 1 represented the fields with surface water or frozen ice and Irrigation 2 represented the fields without surface water but with high soil moisture. The F1-Scores for irrigation 1 on 17 December, 4 January, and 25 January were very high. In the other four dates, two kinds of irrigation condition were classified and the F1-Scores were not kept at the same height. The F1-Scores for irrigation 1 decreased a little and the F1-Scores for Irrigation 2 were in a large range of 72.7 to 95.8. It proves these two types were still able to be separated.

### 4.3. The Irrigated Area Analysis in Watersheds

According to the time series of satellite images, it found that irrigation carried out in early November when quite a few fields were irrigated and unfortunately there was no valid GF-1 satellite image which covered the entire study area available during this period for this study. As it was in winter and the temperature went down to below zero gradually, the irrigated fields were covered by frozen ice due to the cold temperature. The irrigated fields found were increased and suspended in December and January. Ice starts to melt in late-February, and the newly irrigated fields were found again in March and April as it was able to apply irrigation. The largest irrigation area was identified in late-April as the sowing happened in May and the fields must dry up for sowing. Based on these classified images, the irrigation area on each date was calculated for each watershed. Table 6 lists the statistics of irrigation conditions on seven dates for three watersheds in the study area.

**Table 6.** The irrigation area for 3 watersheds (Unit: km$^2$).

|  | **Fen River** | | | **Wenyu River** | | **Xiao River** | |
| --- | --- | --- | --- | --- | --- | --- | --- |
|  | **Sum** | **I1** | **I2** | **I1** | **I2** | **I1** | **I2** |
| 27 December 2022 | 98.6 | 65.8 | - | 22.9 | - | 9.9 | - |
| 4 January 2023 | 166.9 | 115.1 | - | 33.9 | - | 17.9 | - |
| 25 January 2023 | 208.0 | 143.1 | - | 37.6 | - | 27.3 | - |
| 3 March 2023 | 292.8 | 10.2 | 166.1 | 3.4 | 98.5 | 1.1 | 13.5 |
| 27 March 2023 | 538.0 | 6.2 | 306.4 | 1.3 | 166.0 | 0.3 | 57.8 |
| 8 April 2023 | 623.1 | 9.4 | 436.2 | 0.7 | 107.2 | 0.3 | 69.3 |
| 29 April 2023 | 653.8 | 5.8 | 453.1 | 1.1 | 120.5 | 0.9 | 72.4 |

Fen River irrigation area is the largest one in the study area. The area of irrigated fields identified in frozen winter season accounted for 13.7% of total irrigatable land and the area of irrigated fields before sowing increased and accounted for 43.9%. As a tributary of Fen river, Wenyu river irrigable land ranks the second. The area of irrigated fields in winter reached 11.0% of total irrigatable land, and the area of irrigated fields before sowing increased to 35.6%. Xiao river irrigation area is the smallest one. The area of irrigated fields identified in frozen winter season accounted for 12.3% of total irrigatable land and the area of irrigated fields before sowing increased and accounted for 33.0%.

### 5. Discussion

### 5.1. The Challenges of Identifying Irrigation Outside the Growing Season

Our purpose was to know how many and in which fields irrigation has applied before the sowing season in May in spring. Many irrigated fields were able to be retrieved in the classification as the training samples were able to be visually identified. This case study has achieved its original research purpose and may complement the existing methods of mapping irrigation fields in growing season.

However, sometimes, it is not able to make the training samples inclusive. Shallow surface water or soil water in a few irrigated fields evaporates over time and the water in the fields gradually disappears as the air temperature goes up in spring. To distinguish this kind of dry up of irrigated field from other classes becomes indistinct due to the long

interval between two satellite images. These kinds of irrigated fields will be omitted in the classified results as there are no training samples represented in this scenario.

This study was able to identify the irrigated fields but it did not answer which day irrigation was applied and how much water was applied. Both questions were not able to be answered in this study and they should be taken into consideration in the future research.

*5.2. The Consideration in This Irrigation Mapping*

In this study, only 7 scenes of GF-1 images out of growing season were valid and it witnessed the real capacity of GF-1 alone for identifying the irrigation fields. Optical satellite image is prone to cloud contamination. The better the results will be, the more multiple sources satellite images are available. Ideally, if daily and high-quality satellite images are available, it can identify the new irrigation event in time. In this sense, the integration of many more other high resolution satellite data, such as Sentinel-1/2 and Landsat8/9, should improve this study considerably.

In this study area, farmers conduct irrigation to the bare arable land as soon as the winter comes. It is easier to visually identify the irrigated field from bare land than vegetated fields. Two sets of irrigation scenarios in the fields were distinguished. Irrigation 1 represents the fields waterlogged or frozen in winter after the large volume flooding irrigation. Irrigation 2 represents the fields with the high soil moisture but without surface water. Due to the cold temperature and less evaporation in winter, no classification samples for Irrigation 2 were identified on the images of 17 December, 4 January, and 25 January while all irrigation samples represented Irrigation 1 as the irrigated fields were frozen on these dates. In the other four dates, the two kinds of irrigation conditions in the field were able to be identified.

*5.3. The Winter Irrigation Impact on Ecosystem*

The winter irrigation was a kind of cultivation management in the region in order to increase crop yield in the next year. Irrigation out of growing season has the advantage of protecting the ecosystem. It may help reduce the wind erosion due to wet soil in the field surface when the strong wind happens in spring. But a large volume of water applied also brings some adverse ecological effects on the farming system. Sowing in Spring 2023 had to be postponed due to wet soil in the field. On the image of 29 April 2023, it still found surface water on the fields. These fields were not able to be sowed in time. Therefore, the answer to the economic and minimum amount of water put into the field also needs to be further investigated. Soil salinization is another adverse effect induced by irrigation. Large volume of water speeds up evaporation in spring and brings the salt in deep soil back to the field surface. These effects on the ecological system imposed by irrigation out of season are worth further investigating in the near future.

**6. Conclusions**

This study explored the remote sensing-based classification approach to identify irrigated fields out of growing season in the winter season of 2022 to 2023. The proposed classification approach took four spectral bands and all NDVI like indices computed from any two of these four bands of GF-1 satellite data as the input features of the Random Forest algorithm. Regarding the two key parameters of RF, the number of features was set as the square root of the number of input bands of the image while the number of the tree was set to 100. The classification samples corresponding to each image were obtained by visual interpretation with the support of collected field data and then separated into training and validation sets by a ratio of 70% to 30%. Finally, the irrigated fields along with time in Jinzhong basin of Shanxi province, China were retrieved on the seven scenes of valid GF-1 satellite images, respectively.

The results show that the method developed in this study performed well and no overperformance and underperformance were found as the accuracies of classified image were not higher or far lower than that from models. The validations showed that the mean

of the highest out-of-bag accuracies for seven RF models was 94.9% and the mean of the averaged out-of-bag accuracies in the plateau for seven RF models was 94.1%; the overall accuracy for all seven classified outputs was in the range of 86.8–92.5%, Kappa in the range of 84.0–91.0%, and F1-Score in the range of 82.1–90.1%. The lowest OA was 86.8% in comparison with the model accuracy of 92.9%, and the highest OA 92.5% in comparison with the model accuracy of 94.4%. The F1-Scores for irrigation 1 on 17 December, 4 January, and 25 January were very high and in the range of 92.2–97.2%. On the other four dates, the F1-Scores for Irrigation 1 decreased slightly and in the range of 86.0–91.7%, and the F1-Scores for Irrigation 2 were in a large range of 72.7 to 95.8%.

It also found that irrigation in the study area was carried out in early November but the quite few fields started to be irrigated, and the number of irrigated fields increased and suspended in December and January when the irrigated fields were covered by frozen ice and it was not able to apply irrigation due to low temperature. The irrigation was carried out again as the temperature went up in late February. The irrigation extended dramatically in March and April. The largest irrigation area was identified in later April as the sowing happened in May and the fields must dry up for sowing. The area of irrigated fields in the study area were increasing over time with sizes of 98.6, 166.9, 208.0, 292.8, 538.0, 623.1, 653.8 km$^2$ from December to April, accounting for 6.1%, 10.4%, 12.9%,18.2%, 33.4%, 38.7%, and 40.6% of the total irrigatable land in the study area, respectively.

This case study shows that there is another window out of growing season to map the irrigated fields using Random Forest classification algorithm. This knowledge may complement the traditional consideration of retrieving irrigation maps only in growing season with remote sensing images for a large area. It also found too much water was applied in this study area and a few wet fields were not able to be sowed in time. The positive and adverse effect on the ecologic system imposed by irrigation out of season is worth being further investigated in the near future in order to support sustainable water resources management in the region. If the dense and even-distributed time series of valid satellite images may be made available, the irrigated fields over time may be well identified with the proposed approach. It frequently provides better irrigation information to the water resource authority and then the water resource authority may evaluate the excess water usage and its ecological consequences.

**Author Contributions:** Conceptualization, J.F., Q.S. and P.D.; methodology, J.F., Q.S. and Z.Q.; software, J.F. and C.Z.; validation, J.L., W.Z., R.P. and Y.S.; data processing, J.L. and Y.L.; writing, Q.S. and J.F.; comments Z.Q. and P.D. All authors have read and agreed to the published version of the manuscript.

**Funding:** The authors are grateful for the financial support by the National key research and development program(2017YFB0504105), ESA project (Dragon 5 58944).

**Data Availability Statement:** Some or all data, models, or code that support the findings of this study are available from the corresponding author upon reasonable request.

**Acknowledgments:** The authors are grateful for the valuable comments from anonymous reviewers.

**Conflicts of Interest:** The authors declare no conflict of interest.

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
