# Peer review of "Remote Sensing-Based Classification of Winter Irrigation Fields Using the Random Forest Algorithm and GF-1 Data: A Case Study of Jinzhong Basin, North China"

_remotesensing, doi:10.3390/rs15184599_

Round 1
Reviewer 1 Report
Please see the attached file

Acceptable with some needed minor improvements in terms of language and grammar, contextualizing
Author Response
Dear anonymous Reviewer,
First of all, on behalf of our author team, I appreciate you very much for your valuable and constructive comments on our manuscript submitted on Aug 8.2023. Your comments are helping us not only improve our manuscript but also guild us to prepare our future studies and manuscripts.
We have taken all your comments and improved our manuscript in following aspects:
- The language has been polished by our major authors.
- The figure 2b was removed and the Figure1 and 2 were updated.
- Some texts in the manuscript were polished in responding to your questions in the comments.
- Our consideration to your questions were attached below.
Thanks again for your valuable comments
Dr. Jinlong Fan
National Satellite Meteorological Center
China Meteorological Administration

Reviewer 2 Report
The author conducted research on the algorithm of irrigated fields identification out of growing season and developed a remote sensing-based classification approach to identify irrigated fields with Radom Forest algorithm out of growing season. In this paper, the images of seven dates out of growing season are classified separately, and the performance of the seven models is evaluated by OOB. The confusion error matrix with the OA, Kappa and F1-score was used to evaluate the accuracy of the classifications. These results showed that the classification was acceptable and not over performed as the accuracies of all classified images were lower than the models. This case study manifests that there is another window out of growing season to map the irrigated fields. This knowledge may complement the consideration of retrieving irrigation map for a large area. Therefore, I would like to recommend acceptance after minor revision if the authors can explain the following questions.
1. How long is the time span for these irrigation monitoring.
2. What exactly do the 7 models refer to?
3. Why the model accuracy of different date will be different?
4. The category names in Figure 2b appear incomplete. Please consider adjusting them.
5. Why are the I2 values of 2022/12/27, 2023/01/04 and 2023/01/25 absent in Table 6?
Minor editing of English language required.
Author Response
Dear anonymous Reviewer,
First of all, on behalf of our author team, I appreciate you for your valuable and constructive comments on our manuscript submitted on Aug 8.2023.
We have taken all your comments and improved our manuscript in following aspects:
- The figure 2b was removed and the Figure 2 was updated.
- Some texts in the manuscript were polished in considering your questions in the comments.
- Our consideration to your questions were attached below.
Thanks again for your valuable comments
Dr. Jinlong Fan
National Satellite Meteorological Center
China Meteorological Administration
PS: Original Comments and Responses item by item were attached.

Reviewer 3 Report
Overall, this manuscript lacks innovation. The specific suggestions are as follows:
1. The method lacks innovation and does not explain how to relate the time dimension. The section 3.3 is well-known and does not need to be described in the manuscript.
2.The features involved in classification are not targeted. Except for the four bands and vegetation index, there is no targeted mining of other features for classification.
3.The results analysis only listed some pictures and numbers, and the analysis was not thorough.
4. Some parts of the discussion are not relevant to the results analysis, such as lines 349 to 368.
Author Response
Dear anonymous Reviewer,
First of all, on behalf of our author team, I appreciate you for your valuable and constructive comments on our manuscript submitted on Aug 8.2023.
We have taken all your comments and improved our manuscript in following aspects:
- The many texts in the section 3.3 were removed following your suggestions.
- Some texts in the manuscript were polished in considering your questions in the comments.
- Our consideration to your questions were attached below.
Thanks again for your valuable comments
Dr. Jinlong Fan
National Satellite Meteorological Center
China Meteorological Administration

Round 2
Reviewer 3 Report
The research topic of the manuscript is meaningful, but the research methods still lack innovation. The random forest method is a mature method, but it is not omnipotent. The manuscript does not explain why this method is suitable for extracting winter irrigation fields. The manuscript needs to explain the characteristics of winter irrigation fields and how to extract results based on these characteristics.
Author Response
Dear Reviewer,
Our response on your comments is attached in the word document.
Thanks,
Jinlong Fan
